# Associations between serum retinol and all-cause mortality among adults with prediabetes and diabetes: A cohort study

Qing Sun[1], Jie Guo[2,3]*

**1** Department of Traditional Chinese Medicine, Peking Union Medical College Hospital, Chinese Academy of Medical Sciences and Peking Union Medical College, Beijing, China, **2** Department of Neurobiology, Karolinska Institutet, Care Sciences and Society, Stockholm, Sweden, **3** Department of Nutrition and Health, China Agricultural University, Beijing, China

* jie.guo@ki.se

**Data Availability Statement:** All relevant data in this study are publicly available in National Health and Nutrition Examination Survey Homepage (https://www.cdc.gov/nchs/nhanes/index.htm). All-cause mortality was ascertained via linkage with

## Abstract

We aimed to explore the associations between serum retinol and all-cause mortality among people with prediabetes and diabetes. The study included 2582 participants with prediabetes and 1654 with diabetes aged ≥40 years from the National Health and Nutrition Examination Survey 2001–2006. Serum retinol was collected from laboratory tests and categorized into five groups, including <50, 50–60, 60–70, 70–80, and ≥80 μg/dL. Deaths were obtained by linkage to National Death Index up to December 31, 2019. Cox proportional hazards models were used to estimate the associations between serum retinol and all-cause mortality. During the follow-up, 993 participants with prediabetes died and 874 participants with diabetes died. There were U-shaped associations between serum retinol and mortality among participants with prediabetes and diabetes, separately. Among participants with prediabetes, compared to serum retinol levels of 50–60 μg/dL, the hazard ratio (HR) (95% confidence interval [CI]) of mortality was 1.40 (95% CI 1.11 to 1.76) and 1.26 (95% CI 1.00 to 1.57) for serum retinol <50 or ≥80 μg/dL, respectively. Among participants with diabetes, compared to serum retinol levels of 50–60 μg/dL, the hazard ratio (HR) (95% confidence interval [CI]) of mortality was 1.25 (95% CI 0.96 to 1.62) and 1.21 (95% CI 0.91 to 1.62) for serum retinol <50 or ≥80 μg/dL, respectively. The U-shaped associations between serum retinol and mortality still existed among participants aged ≥60 years with prediabetes or diabetes but were not statistically significant among those aged 40–59 years with prediabetes or diabetes. In conclusion, both low and excessive serum retinol tended to be with higher mortality risk among people with abnormal blood glucose.

## Introduction

Diabetes is a major global public health problem, affecting an estimated 463 million adults [1]. Moreover, about 374 million people had prediabetes worldwide in 2019 [1]. Both diabetes and prediabetes have been reported to be with increased risk of death [2,3].

the National Death Index (https://www.cdc.gov/nchs/ndi/index.htm).

**Funding:** J.G. received grants from the China Scholarship Council (No. 201808340062). The funders had no role in study design, data collection and analysis, decision to publish, or preparation of the manuscript.

**Competing interests:** The authors have declared that no competing interests exist.

Vitamin A (VA) is a class of essential, fat-soluble compounds that primarily circulates as retinol—the biochemical indicator of human VA status [4]. Humans are unable to synthesize VA but can obtain it from both animal food (e.g., milk, liver, cheese, butter, and egg) and plant-based pro-vitamin A carotenoids [5,6]. VA plays a crucial role in regulating cellular differentiation, vision, and immunity, and the deficiency of VA has been linked to the onset and development of metabolic disorders, such as diabetes [7,8]. However, the associations between serum retinol and mortality from population-based studies were inconsistent. Some studies found that serum retinol was inversely associated with all-cause mortality [9,10], while others found no significant association between serum retinol and all-cause mortality [11–13]. Nevertheless, two studies based on the Third National Health and Nutrition Examination Survey (NHANES III) database reported a U-shaped association between serum retinol and mortality [14,15]. Moreover, there is a lack of evidence regarding the associations between serum retinol and the prognosis of adults with prediabetes and diabetes, particularly concerning mortality.

We hypothesized that both extremely low and high levels of serum retinol may be associated with increased mortality among adults with prediabetes or diabetes. To verify these hypotheses, we aimed to investigate the associations between serum retinol and all-cause mortality based on a nationally representative sample of United States (U.S.) adults with prediabetes or diabetes.

## Materials and methods

### Study population

The NHANES is an ongoing population-based project performed in the national representative population of the U.S. [16]. In this study, we included six survey cycles from 2001–2002 to 2005–2006, which collected information about serum retinol. Among 4490 participants aged ≥40 years who were not pregnant or lactating and with prediabetes or diabetes, we further excluded 254 with missing information about serum retinol. Finally, we included 2582 participants with prediabetes (i.e., glycated hemoglobin A1c [HbA1c] ranging from 5.7 to 6.4% or fasting plasma glucose [FPG] ranging from 100 to 125 mg/dL), and 1654 with diabetes (i.e., self-reported diagnosis, FPG ≥126 mg/dL, HbA1c ≥6.5%, or use of insulin or oral hypoglycemic medications) (**S1 Fig in S1 File**).

The Ethics Review Board for the National Center for Health approved the NHANES data collection and allowed the data files for public use. Every participant provided written informed consent before data collection. The authors had no access to information that could identify individual participants during or after data collection.

### Assessment of serum retinol

Serum retinol was collected from laboratory tests using high-performance liquid chromatography with photodiode array detection. Detailed information for laboratory procedures is under NHANES databases (https://wwwn.cdc.gov/nchs/nhanes/Default.aspx).

We considered serum retinol as both continuous variables and categorical variables (Five groups: <50, 50–60, 60–70, 70–80, and ≥80 µg/dL).

### Assessment of mortality

All-cause mortality was ascertained via linkage with the National Death Index (NDI) up to December 31, 2019. Cardiovascular disease (CVD) death was defined as death from diabetes mellitus (E10-E14), heart disease (codes I00-I09, I11, I13, and I20-I51), or cerebrovascular disease (codes I60-I69). Non-CVD death was defined as death from all other causes. Detailed

information about the methodology to link the NHANES data to the NDI has been described elsewhere [17,18].

## Assessment of covariates

Information on sociodemographic factors (age, sex, ethnicity, and education), lifestyle factors (smoking status, alcohol consumption, and physical activity), medical conditions (hypertension and diabetes), use of prescription medications (antihypertensive treatment and glucose-lowering medications) were collected through the computer-assisted questionnaire. Blood pressure, weight, height, and waist circumference were measured by trained staff. HbA1c, FPG, and total cholesterol were collected from laboratory tests of blood specimens.

Age was categorized into 40–59 vs. $\geq$60 years. Ethnicity includes non-Hispanic white, non-Hispanic black, Mexican American, and other (other Hispanic and other race—including multi-racial). Education was grouped based on the highest degree received as less than high school, high school, and more than high school. Smoking status was categorized as current smoker, former smoker, and non-smoker. Alcohol consumption was dichotomized as drinker if participants self-reported that they had at least 12 alcoholic drinks per year or non-drinker if they had less than 12 alcoholic drinks per year. Physical activity was dichotomized into active ($\geq$150 mins per week for moderate-intensity or $\geq$75 mins per week for vigorous-intensity activity) and inactive. Body mass index (BMI) was calculated using weight (kg) divided by squared height (m$^2$). Prediabetes was defined as HbA1c ranging from 5.7 to 6.4% or FPG ranging from 100 to 125 mg/dL. Diabetes was determined based on self-reported medical history, or use of glucose-lowering medications, or FPG $\geq$126 mg/dL, or HbA1c $\geq$6.5% [19]. Hypertension was defined based on systolic and diastolic blood pressure (SBP/DBP) $\geq$140/90 mmHg, self-reported medical history, or self-report of taking antihypertensive treatment.

## Statistical analysis

The NHANES survey design and 6-year sampling weights were incorporated into all statistical analyses in this study [20].

To account for both the event onset and the time to the event, we calculated hazard ratios (HRs) and 95% confidence intervals (CIs) for the associations of serum retinol (as a continuous or categorical variable) with all-cause mortality using Cox proportional hazards models. Schoenfeld residuals were calculated to test the proportional hazard assumption, resulting in no violations. We used follow-up time as the timescale, calculated as the time from the initial date of the NHANES survey cycle until the date of death or December 31, 2019, whichever occurred first. To minimize the effect of potential confounders on the associations between serum retinol and mortality, we adjusted for age (continuous), sex (female or male), and ethnicity (non-Hispanic white, non-Hispanic black, Mexican American, or other); and subsequently adjusted for education (less than high school, high school or equivalent, or college or above), smoking status (never smoker, former smoker, or current smoker), alcohol consumption (drinker or non-drinker), physical activity (inactive or active), total cholesterol (continuous), and hypertension (no or yes). We first explored the associations of continuous serum retinol with mortality using the restricted cubic spline with four knots at the 5th, 35th, 65th, and 95th percentiles. We further categorized participants using the turning point of serum retinol identified from the restricted cubic spline and analyzed the associations between continuous serum retinol (per 10 μg/dL increase) and mortality separately. We then assessed the associations between categorical serum retinol in five groups and mortality. The multiplicative interaction between age groups and serum retinol groups for the mortality were also assessed by including their cross-product term in the model and further conducted stratified analyses by age groups (40–59 and $\geq$ 60 years).

In the supplementary analyses, we 1) adjusted waist circumference instead of BMI; 2) assessed the associations of serum retinol with CVD and non-CVD death. All statistical analyses were performed using SAS software (version 9.4). All *P* values were two-sided, and statistical significance was defined as *P* <0.05. Weighted percentages and means (standard error [SE]) were calculated using SURVEYFREQ or SURVEYMEANS procedure, respectively. The Bonferroni correction method was used for multiple comparisons and multiple tests. SURVEYPHREG procedures were used to calculate the associations between serum retinol and mortality. Data were analyzed from April 2022 to December 2022.

## Results

### Baseline characteristics of the study population

Among participants with prediabetes, those with higher serum retinol were older, more likely to be male, to be non-Hispanic White, to be with a higher education, to be drinker, and to have a higher prevalence of hypertension, a higher level of total cholesterol and a lower level of BMI (*P* <0.05 for all) (**Table 1**).

Among participants with diabetes, those with higher serum retinol were older, more likely to be physically active, and to have a higher prevalence of hypertension, a higher level of total cholesterol, and a lower level of BMI (*P* <0.05 for all) (**Table 1**).

### Association of serum retinol with mortality

During a median follow-up of 14.3 years, 993 participants with prediabetes died (34.1% died from cardiovascular diseases and 65.9% died from non-cardiovascular diseases). There was a U-shaped association between serum retinol and mortality, with a lowest risk around the level of 68.5 ug/dL (**Fig 1**). Among participants with prediabetes and with serum retinol <68.5 μg/dL, the multi-adjusted HR per 10 μg/dL increase in serum retinol was 0.82 (95% CI 0.72 to 0.94) (**Table 2**). Among those with serum retinol ≥68.5 μg/dL, the multi-adjusted HR per 10 μg/dL increase in serum retinol was 1.20 (95% CI 1.08 to 1.33). When using the serum retinol as a continuous variable, compared to participants with serum retinol levels of 50–60 μg/dL, the multi-adjusted HR (95% CI) of mortality was 1.40 (95% CI 1.11 to 1.76) and 1.26 (95% CI 1.00 to 1.57) for those with serum retinol <50 or ≥80 μg/dL, respectively.

During a median follow-up of 13.5 years, 874 participants with diabetes died (44.9% died from cardiovascular diseases and 55.1% died from non-cardiovascular diseases). There was a U-shaped association between serum retinol and mortality among participants with diabetes at baseline, with a lowest risk around the level of 58.5 ug/dL (**Fig 1**). Among participants with diabetes and with serum retinol <58.5 μg/dL, the multi-adjusted HR per 10 μg/dL increase in serum retinol was 0.76 (95% CI 0.64 to 0.90). Among those with serum retinol ≥58.5 μg/dL, the multi-adjusted HR per 10 μg/dL increase in serum retinol was 1.06 (95% CI 0.98 to 1.14). Compared to participants with serum retinol levels of 50–60 μg/dL, those with extremely low or high level of serum retinol <50 (1.25 [0.96 to 1.62]) or ≥80 μg/dL (1.21 [0.91 to 1.62]) tended to be with increased mortality albeit non-significant (**Table 2**).

### Association of serum retinol with mortality by age groups

The associations between serum retinol and mortality tended to be U-shaped among participants with prediabetes or diabetes across age groups (40–59 years and ≥60 years) (**S2 Fig in S1 File**). Among participants aged 40–59 years with diabetes, compared to participants with serum retinol 50–60 ug/dL, the risk of mortality for those with serum retinol ≥ 80 μg/dL was higher (HR 2.23, 95% CI 1.19 to 4.17) (**Table 3**). Among participants with prediabetes aged

**Table 1. Baseline characteristics of the study population with prediabetes and diabetes by serum retinol levels.**

| Characteristic | Among participants with prediabetes | | | | | Among participants with diabetes | | | | |
|---|---|---|---|---|---|---|---|---|---|---|
| | <50 µg/dL | 50–60 µg/dL | 60–70 µg/dL | 70–80 µg/dL | ≥80 µg/dL | <50 µg/dL | 50–60 µg/dL | 60–70 µg/dL | 70–80 µg/dL | ≥80 µg/dL |
| **Age group, years** | 56.23 (0.86) | 57.91 (0.65) | 59.31 (0.59) | 60.25 (0.78) | 62.29 (0.68) | 57.13 (0.75) | 59.96 (0.88) | 63.56 (0.77) | 62.95 (1.08) | 64.82 (0.92) |
| <60 | 67.7 | 60.4 | 56.0 | 51.7 | 45.7 | 61.9 | 52.4 | 38.9 | 39.2 | 32.4 |
| ≥60 | 32.3 | 39.6 | 44.0 | 48.3 | 54.3 | 38.1 | 47.6 | 61.1 | 60.8 | 67.6 |
| **Sex** | | | | | | | | | | |
| Male | 35.2 | 51.8 | 54.3 | 60.4 | 61.8 | 41.5 | 55.2 | 50.7 | 54.0 | 54.3 |
| Female | 64.8 | 48.2 | 45.7 | 39.6 | 38.2 | 58.5 | 44.8 | 49.3 | 46.0 | 45.7 |
| **Ethnicity** | | | | | | | | | | |
| Mexican American | 8.8 | 6.2 | 4.2 | 3.2 | 2.2 | 13.3 | 8.3 | 6.3 | 5.2 | 3.8 |
| other | 11.0 | 11.7 | 7.4 | 8.5 | 6.4 | 12.8 | 7.7 | 12.9 | 8.6 | 10.2 |
| Non-Hispanic White | 59.7 | 72.3 | 77.3 | 81.7 | 84.9 | 55.5 | 69.0 | 66.7 | 75.2 | 73.3 |
| Non-Hispanic Black | 20.6 | 9.7 | 11.2 | 6.6 | 6.5 | 18.4 | 15.1 | 14.1 | 11.0 | 12.7 |
| **Education group** | | | | | | | | | | |
| Less than high school | 29.1 | 22.4 | 19.0 | 17.3 | 17.3 | 35.4 | 25.9 | 29.5 | 23.6 | 27.8 |
| High school | 28.1 | 24.3 | 27.4 | 26.4 | 28.1 | 23.4 | 24.4 | 28.5 | 25.7 | 30.3 |
| More than high school | 42.8 | 53.2 | 53.6 | 56.3 | 54.6 | 41.2 | 49.7 | 42.0 | 50.7 | 41.9 |
| **Smoking status** | | | | | | | | | | |
| Current | 30.0 | 21.7 | 21.8 | 17.5 | 14.4 | 25.2 | 19.0 | 19.2 | 13.0 | 12.5 |
| Former | 23.9 | 32.4 | 33.2 | 37.1 | 39.7 | 30.3 | 33.6 | 35.2 | 42.4 | 39.0 |
| Never | 46.2 | 45.8 | 45.0 | 45.4 | 45.9 | 44.5 | 47.4 | 45.6 | 44.5 | 48.5 |
| **Alcohol consumption** | | | | | | | | | | |
| Non-drinker | 41.5 | 36.4 | 29.8 | 27.5 | 25.8 | 35.1 | 40.9 | 38.2 | 42.5 | 44.4 |
| Drinker | 58.5 | 63.6 | 70.2 | 72.5 | 74.2 | 64.9 | 59.1 | 61.8 | 57.5 | 55.6 |
| **Physical activity** | | | | | | | | | | |
| Inactive | 72.3 | 63.3 | 65.2 | 60.5 | 61.6 | 80.6 | 74.0 | 70.1 | 71.8 | 64.1 |
| Active | 27.7 | 36.7 | 34.8 | 39.5 | 38.4 | 19.4 | 26.0 | 29.9 | 28.2 | 35.9 |
| **Hypertension** | 43.7 | 45.4 | 53.7 | 54.9 | 60.8 | 61.4 | 64.7 | 72.4 | 74.3 | 81.6 |
| **TC, mg/dl** | 196.23 (2.19) | 210.38 (2.13) | 213.59 (2.48) | 218.88 (2.81) | 216.19 (2.58) | 193.83 (2.09) | 195.25 (3.26) | 194.12 (2.84) | 207.91 (3.57) | 211.93 (4.95) |
| **BMI, kg/m²** | 31.24 (0.44) | 30.52 (0.42) | 29.47 (0.28) | 29.97 (0.37) | 28.62 (0.29) | 33.63 (0.61) | 32.55 (0.69) | 31.92 (0.55) | 30.58 (0.43) | 31.42 (0.51) |

TC, total cholesterol; BMI, body mass index.

Data are presented as weighted mean (standard error) for continuous variables and weighted percentages for categorical variables.

≥60 years, compared to participants with serum retinol 50–60 µg/dL, the risk of mortality for those with serum retinol<50 ug/dL was higher (HR 1.39, 95% CI 1.06 to 1.83) (**Table 3**). Among those with diabetes aged ≥60 years, compared to participants with serum retinol 50–60 µg/dL, the risk of mortality for those with serum retinol<50 ug/dL was higher (HR 1.46, 95% CI 1.03 to 2.07). The multiplicative interaction between age groups (40–59 vs. ≥60 years) and serum retinol groups on mortality was not statistically significant among participants with prediabetes ($P = 0.527$) or diabetes ($P = 0.085$).

## Supplementary results

In the multi-adjusted model, we included waist circumference instead of BMI and the results were not altered largely (**S1 Table in S1 File**). For cause-specific mortality, among participants

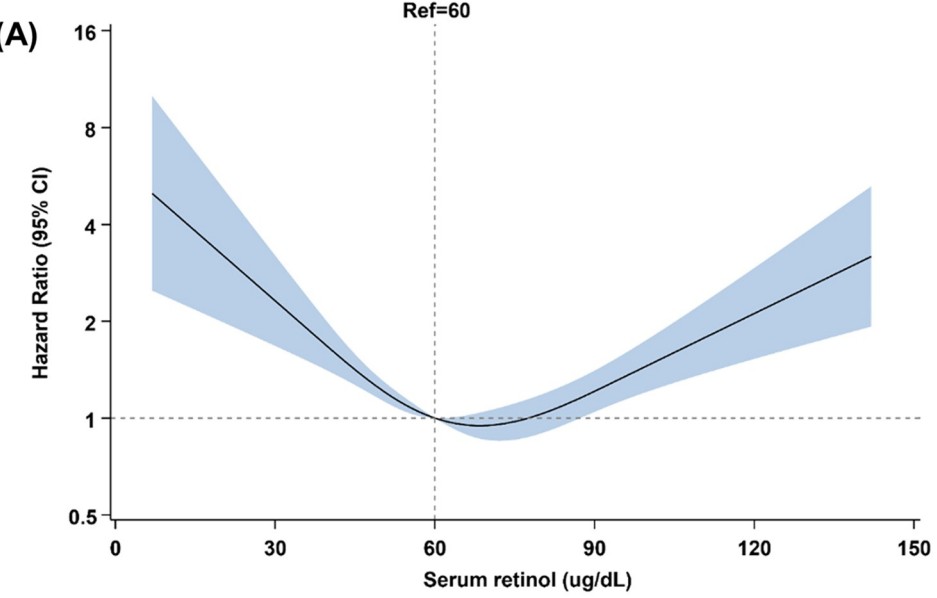

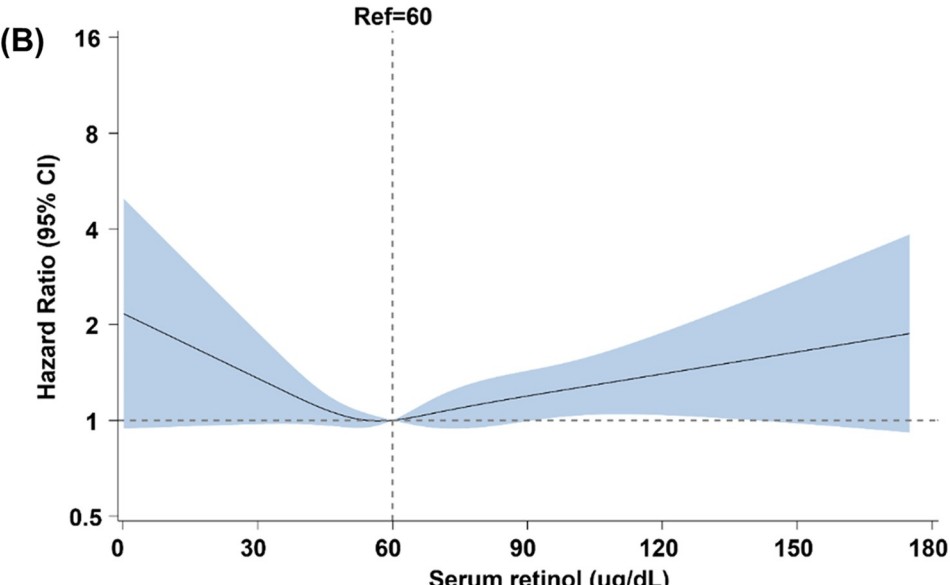

**Fig 1.** The association between serum retinol (μg/dL) and all-cause mortality among participants with prediabetes (A) and diabetes (B) separately. Adjusted for age, sex, ethnicity, education, smoking status, alcohol consumption, physical activity, body mass index, hypertension, total cholesterol. P-values for overall association and P-values for nonlinear association were both < .05 among participants with prediabetes or diabetes.

with prediabetes, extremely level of serum retinol tended to be associated with increased risk of CVD and non-CVD mortality albeit not significant (**S2 Table in S1 File**). Among participants with diabetes, a lower level of serum retinol (<50 μg/dL) was associated with increased non-CVD mortality.

**Table 2. Hazard ratios (HR) and 95% confidence intervals (CI) for all-cause mortality by serum retinol among prediabetes and diabetes separately.**

| Serum retinol | No. of participants | Person years | No. of events | HR (95% CI) [a] | HR (95% CI) [b] |
|---|---|---|---|---|---|
| **Among participants with prediabetes** | | | | | |
| Continuous, per 10 μg/dL increase | | | | | |
| <68.5 μg/dL [c] | 1716 | 22389 | 636 | 0.79 (0.70–0.88)* | 0.82 (0.72–0.94)* |
| ≥68.5 μg/dL [c] | 866 | 11135 | 357 | 1.16 (1.06–1.28)* | 1.20 (1.08–1.33)* |
| Categories | | | | | |
| <50 μg/dL | 592 | 7600 | 215 | 1.46 (1.21–1.76)* | 1.40 (1.11–1.76)* |
| 50–60 μg/dL | 633 | 8425 | 224 | Reference | Reference |
| 60–70 μg/dL | 584 | 7604 | 227 | 0.95 (0.78–1.15) | 0.98 (0.80–1.21) |
| 70–80 μg/dL | 391 | 5282 | 142 | 0.81 (0.67–0.99)* | 0.86 (0.67–1.12) |
| 80+ μg/dL | 382 | 4612 | 185 | 1.06 (0.87–1.28) | 1.26 (1.00–1.57)* |
| **Among participants with diabetes** | | | | | |
| Continuous, per 10 ug/dL increase | | | | | |
| <58.5 μg/dL [d] | 709 | 8733 | 321 | 0.67 (0.57–0.78)* | 0.76 (0.64–0.90)* |
| ≥58.5 μg/dL [d] | 945 | 10612 | 553 | 1.04 (0.98–1.11) | 1.06 (0.98–1.14) |
| Categories | | | | | |
| <50 μg/dL | 401 | 4879 | 176 | 1.43 (1.09–1.87)* | 1.25 (0.96–1.62) |
| 50–60 μg/dL | 370 | 4618 | 178 | Reference | Reference |
| 60–70 μg/dL | 336 | 3814 | 183 | 1.14 (0.86–1.50) | 1.10 (0.81–1.49) |
| 70–80 μg/dL | 238 | 2787 | 134 | 1.00 (0.75–1.34) | 0.96 (0.69–1.34) |
| 80+ μg/dL | 309 | 3248 | 203 | 1.22 (0.95–1.56) | 1.21 (0.91–1.62) |

[a] Adjusted for age, sex, and ethnicity

[b] Further adjusted for education, smoking status, alcohol consumption, physical activity, BMI, hypertension, total cholesterol.

[c] The level of 68.5 ug/dL was the value of serum retinol associated with the lowest risk of mortality among participants with prediabetes.

[d] The level of 58.5 ug/dL was the value of serum retinol associated with the lowest risk of mortality among participants with diabetes.

* $P <0.05$.

## Discussion

In this large prospective cohort study of U.S. adults aged ≥40 years, we found a U-shaped association between serum retinol and all-cause mortality among participants with prediabetes or diabetes. The nadir of the curve for serum retinol and mortality was around 70 μg/dL for participants with prediabetes and 60 μg/dL for those with diabetes. Among older adults aged ≥60 years with prediabetes or diabetes, those with relatively low serum retinol (i.e., <50 μg/dL) had increased risk of mortality.

Previous research has investigated the link between serum retinol and mortality risk in the general population, but the results have been inconsistent. While some studies have found no significant associations [11–13], others have observed negative associations [9,10], and some have reported U-shaped associations [14,15]. In a study with 441 well-nourished older Australian adults (45% with type 2 diabetes), serum retinol was inversely related to 5-year CVD mortality (1). Among 29,104 male smokers in southwestern Finland, 23,797 deaths were identified during a 30-year follow-up, and participants with higher serum retinol experienced significantly lower overall mortality [10]. Interestingly, others have shown no significant associations [11–13]. Among 3254 Japanese inhabitants aged 39–85 years, no obvious relationship has been documented between serum retinol and CVD mortality [12]. However, in the NHANES III Study with 16,008 participants aged 20 years and above, there was a U-shaped association between serum retinol and total mortality, and participants with concentrations in either the

**Table 3. Hazard ratio (HR) and 95% confidence interval (CI) for all-cause mortality of serum retinol among adults across age groups.**

| Serum retinol | No. of participants | Person years | No. of events | HR (95% CI) [a] | HR (95% CI) [b] |
|---|---|---|---|---|---|
| **Aged 40 to 59 years** | | | | | |
| **Among participants with prediabetes** | | | | | |
| <50 µg/dL | 300 | 4435 | 36 | 1.49 (0.77–2.89) | 1.19 (0.57–2.51) |
| 50–60 µg/dL | 278 | 4248 | 27 | Reference | Reference |
| 60–70 µg/dL | 228 | 3423 | 26 | 1.13 (0.61–2.08) | 1.18 (0.56–2.52) |
| 70–80 µg/dL | 147 | 2326 | 16 | 1.05 (0.52–2.12) | 1.03 (0.43–2.50) |
| 80+ µg/dL | 119 | 1810 | 17 | 1.16 (0.56–2.42) | 1.46 (0.60–3.53) |
| **Among participants with diabetes** | | | | | |
| <50 µg/dL | 177 | 2565 | 38 | 1.41 (0.76–2.61) | 0.95 (0.45–1.98) |
| 50–60 µg/dL | 131 | 1943 | 25 | Reference | Reference |
| 60–70 µg/dL | 85 | 1158 | 25 | 1.71 (0.88–3.31) | 1.80 (0.84–3.86) |
| 70–80 µg/dL | 63 | 904 | 18 | 1.17 (0.46–2.94) | 0.92 (0.28–3.05) |
| 80+ µg/dL | 66 | 870 | 22 | 1.61 (0.84–3.07) | 2.23 (1.19–4.17)* |
| **Aged ≥60 years** | | | | | |
| **Among participants with prediabetes** | | | | | |
| <50 µg/dL | 292 | 3165 | 179 | 1.41 (1.08–1.84)* | 1.39 (1.06–1.83)* |
| 50–60 µg/dL | 355 | 4177 | 197 | Reference | Reference |
| 60–70 µg/dL | 356 | 4181 | 201 | 0.9 (0.73–1.10) | 0.92 (0.74–1.14) |
| 70–80 µg/dL | 244 | 2957 | 126 | 0.77 (0.65–0.93)* | 0.84 (0.67–1.05) |
| 80+ µg/dL | 263 | 2802 | 168 | 1.02 (0.81–1.27) | 1.19 (0.96–1.49) |
| **Among participants with diabetes** | | | | | |
| <50 µg/dL | 224 | 2313 | 138 | 1.42 (1.05–1.93)* | 1.46 (1.03–2.07)* |
| 50–60 µg/dL | 239 | 2675 | 153 | Reference | Reference |
| 60–70 µg/dL | 251 | 2656 | 158 | 1.00 (0.77–1.30) | 0.97 (0.72–1.31) |
| 70–80 µg/dL | 175 | 1883 | 116 | 0.95 (0.73–1.24) | 0.95 (0.72–1.27) |
| 80+ µg/dL | 243 | 2378 | 181 | 1.13 (0.86–1.49) | 1.12 (0.82–1.52) |

[a] Adjusted for age, sex, and ethnicity

[b] Further adjusted for education, smoking status, alcohol consumption, physical activity, BMI, hypertension, total cholesterol.

* $P < 0.05$.

quintile 3 or 4 categories (but not in quintile 5) experienced 18% reduced total mortality compared with those in the lowest quintile (HR [95% CI] = 0.82 [0.68, 0.98] and 0.82 [0.69, 0.97], respectively; P trend = 0.93) [14]. In another study based on the NHANES III Study with 6,069 participants aged 50 years or older, there was a U-shaped association and participants with deficient (<30 µg/dL; HR [95% CI] = 2.9 [2.0, 4.2]) or excessive (>80 µg/dL; HR [95% CI] = 1.2 [1.1, 1.4]) retinol had higher mortality risk compared to those with normal retinol concentrations (30–80 µg/dL) [15]. Such inconsistent results may be due to differences in population characteristics (gender, age, ethnicity, comorbidities, etc.), follow-up time, or study outcomes. Moreover, evidence about the association between serum retinol and mortality among people with abnormal blood glucose is lacking. In the current study, we found that among people with prediabetes or diabetes, both extremely low and high serum retinol tended to be associated with higher all-cause mortality. Additionally, we observed U-shaped associations between serum retinol levels and all-cause mortality in the population of older adults aged 60 years and above who had prediabetes or diabetes. Particularly, those with relatively low serum retinol levels (i.e., less than 50 µg/dL) were found to have a significantly increased risk of all-cause mortality.

There are several plausible pathophysiological mechanisms supporting the U-shaped association between serum retinol and all-cause mortality. Retinol plays an important role in regulating cellular differentiation, immunity, and vision [4–6]. Moreover, some in vitro studies have reported that lower levels of retinoids (the derivative of retinol) may be associated with increased oxidative stress, inflammation, and endothelial dysfunction [21,22]. Those, in turn, may lead to higher mortality risk. More importantly, we found that the relatively low serum retinol (i.e., <50 μg/dL) was associated with increased risk of mortality in older adults aged ≥60 years. We speculate that aging is especially vulnerable to malnutrition, and more VA is needed to improve the nutritional status in elderly people. On the other hand, there may be a direct acute and chronic toxicity of hypervitaminosis A [23]. Elevated retinol levels have been related to embryonic malformations and chronic toxicity, such as ataxia, alopecia, hyperlipidemia, hepatotoxicity, fracture, and visual impairments [24,25]. Furthermore, high retinol levels have been documented as an additional oxidative damage factor [26]. In addition, retinol and its derivatives may impact mitochondrial structure and function by causing organelle swelling [27], which could worsen bioenergy disorders, increase oxidative stress, and apoptosis or necrosis. Moreover, serum retinol can bind to retinol-binding protein 4 (RBP4), and is delivered to adipose tissues, further stimulate macrophages bringing about increased local inflammation and systemic insulin resistance through a JNK-dependent pathway [28]. Although relatively scarce, some evidence has shown that VA status and its metabolism may contribute to the glucose regulation and fat metabolism [7,29]. Moderate retinoic acid treatment has been shown to reduce blood glucose in Zucker diabetic fatty (ZDF) rats [30], whereas excessive VA intake impaired glucose tolerance and fatty liver in ZDF rats [31,32]. These results may be due to the synergistic interactions of insulin and retinoic acid through the activations retinoid X receptors and retinoic acid receptors signaling pathways [33].

Overall, our findings are just hypothesis-generating and more studies are needed to verify our findings and to explore the underlying mechanisms. The strengths of the current study include the prospective study design, a relatively large sample size, and a nationally representative sample with prediabetes and diabetes. However, several limitations should be noted. First, we could not conclude a causal association between serum retinol and mortality because of the observational study design. Second, serum retinol may change over time. However, only a single time-point serum retinol concentration was available in the current study, and its changes over time might lead to a misclassification of the serum retinol groups. More studies with longitudinal serum retinol measurements are warranted to understand the associations between serum retinol and mortality. Third, mortality outcomes were determined by a probabilistic match to the NDI, which could lead to misclassification. However, a previous validation study demonstrated a high accuracy of this method [34]. Finally, although we adjusted a wide range of confounders, there may be residual confounding effects due to not adjusting for potential confounders (e.g., vitamin A supplements, nutritional diet, duration of diabetes, and diabetes complications). Therefore, more studies, including intervention trials, are needed to clarify the risks or benefits of serum retinol levels on the mortality risk among prediabetes and diabetes.

## Conclusions

Our study found a U-shaped relationship between serum retinol and all-cause mortality among participants aged ≥40 years with prediabetes or diabetes. The nadir of the curve for serum retinol and mortality was around 70 μg/dL and 60 μg/dL among participants with prediabetes and diabetes, respectively. Our findings suggested that managing serum retinol within the optimal range may reduce the mortality risk in people with prediabetes or diabetes. Further studies are warranted to confirm whether VA supplements may improve serum retinol among people with abnormal blood glucose and with relatively low retinol level.

## Supporting information

**S1 Checklist. PLOSOne clinical studies checklist.**
(DOCX)

**S2 Checklist. STROBE checklist cohort.**
(DOCX)

**S1 File. Supplementary file.**
(DOCX)

## Acknowledgments

The author appreciates the diligent efforts of researchers involved in the National Health and Nutrition Examination Surveys. The findings and conclusions in this report are those of the author and do not necessarily represent the official position of the Centers for Disease Control and Prevention.

## Author Contributions

**Conceptualization:** Qing Sun.

**Data curation:** Jie Guo.

**Formal analysis:** Jie Guo.

**Funding acquisition:** Jie Guo.

**Investigation:** Jie Guo.

**Methodology:** Jie Guo.

**Project administration:** Jie Guo.

**Resources:** Jie Guo.

**Software:** Jie Guo.

**Supervision:** Jie Guo.

**Validation:** Jie Guo.

**Visualization:** Jie Guo.

**Writing – original draft:** Qing Sun.

**Writing – review & editing:** Qing Sun.

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
