## [Decision Letter · Decision Letter 0]

24 Nov 2023

PONE-D-23-19123Associations between serum retinol and all-cause mortality among adults with prediabetes and diabetes: A cohort studyPLOS ONE

Dear Dr. Guo,

Thank you for submitting your manuscript to PLOS ONE. After careful consideration, we feel that it has merit but does not fully meet PLOS ONE’s publication criteria as it currently stands. Therefore, we invite you to submit a revised version of the manuscript that addresses the points raised during the review process.

We look forward to receiving your revised manuscript.

Kind regards,

Billy Morara Tsima, MD MSc

Academic Editor

PLOS ONE

Journal Requirements:

Important: If there are ethical or legal restrictions to sharing your data publicly, please explain these restrictions in detail. Please see our guidelines for more information on what we consider unacceptable restrictions to publicly sharing data: http://journals.plos.org/plosone/s/data-availability#loc- unacceptable-data-access-restrictions. Note that it is not acceptable for the authors to be the sole named individuals responsible for ensuring data access.

Reviewers' comments:

Reviewer's Responses to Questions

**Comments to the Author**

1. Is the manuscript technically sound, and do the data support the conclusions?

Reviewer #1: Yes

Reviewer #2: Partly

2. Has the statistical analysis been performed appropriately and rigorously? 

Reviewer #1: Yes

Reviewer #2: No

3. Have the authors made all data underlying the findings in their manuscript fully available?

Reviewer #1: Yes

Reviewer #2: No

4. Is the manuscript presented in an intelligible fashion and written in standard English?

Reviewer #1: No

Reviewer #2: Yes

5. Review Comments to the Author

Reviewer #1: Dear Authors,

This research addresses a relevant subject considering previous studies that support the relationship between retinol levels with diabetes and all-cause mortality. Moreover, reinforce the need for research designed to fill the knowledge gaps that may impact the approach of such individuals related to retinol levels. The manuscript deserves minor revision due to some grammar mistakes.

Yours Sincerely

Reviewer #2: The manuscript, the title of which is Associations between serum retinol and all-cause mortality among adults with

prediabetes and diabetes: A cohort study, is very interesting and intriguing.

1)About the writing of the manuscript, even though it is written in standard English, it is very confusing. The manuscript should be rewritten according to a clearer and more logical manner. A dramatic revision is mandatory for making this manuscript crystal clear for the potential readers. A lot of results with Confidence Intervals are provided without a logical order. The manuscript is extremely confusing for any reviewer highly specialized in statistical methods. For the common reader, reading this manuscript as it has been submitted will cause headaches.

2) The tables are far away from the text and are not easy to read. Additionally, the legends of these tables should help any common reader to read more easily and understand without any doubt the tables.

3) The Material and Methods section is quite short and does not contain any crystal clear explanation justifying the statistical method used.

4) Are prediabetes patients receiving any medication or not ? Are they following any nutritional diet or not ? If they receive any medication or diet, what are these medications or diet(s) ?

5) Usually Diabetes Mellitus (DM) provokes complications such as the diabetic foot, the Diabetic retinopathy, the Diabetic nephropathy and can provoke strokes and other cardio-vascular complications alone or in association with smoking ? Ths manuscript does not provide the causes of death of ant patient enrolled in this study.

6) The BMI tends to be replaced by the perimeter of the waist.

7) The duration of DM from its onset to the appearance of complications is a parameter of paramount importance in the prediction of complications including death

8) The association of Hypertension with DM can actually lead to severe metabolic syndromes : The manuscript does not evoke how many metabolic syndromes were observed in the cohort studied.

9) No mention of the monitoring of the patients enrolled is provided. How many patients were followed by a cardiologist, by a Nephrologist ? Monitoring the heart of the patients by ultrasonography-Doppler, monitoring the brain of each patient by MRI should have been done in order to evaluate the macro vascular damages caused by DM.

10) The rationale for studying retinol is unclear. What is (are) the pathophysiological mechanism(s) susceptible to explain a higher mortality for patients having low concentrations of retinol or high concentrations of retinol in their blood. The answer to this request is of utmost importance for the reviewers and the potential readers.

6. PLOS authors have the option to publish the peer review history of their article (what does this mean?). If published, this will include your full peer review and any attached files.

Reviewer #1: No

Reviewer #2: No

---

## [Author Response · Author response to Decision Letter 0]

25 Dec 2023

Ms. Ref. No.: PONE-D-23-19123

Reviewer #1: Dear Authors,

This research addresses a relevant subject considering previous studies that support the relationship between retinol levels with diabetes and all-cause mortality. Moreover, reinforce the need for research designed to fill the knowledge gaps that may impact the approach of such individuals related to retinol levels. The manuscript deserves minor revision due to some grammar mistakes.

We thank the reviewer for the helpful comments. Per the reviewer’s suggestion, we have checked and revised the language carefully.

Reviewer #2: The manuscript, the title of which is Associations between serum retinol and all-cause mortality among adults with prediabetes and diabetes: A cohort study, is very interesting and intriguing.

1)About the writing of the manuscript, even though it is written in standard English, it is very confusing. The manuscript should be rewritten according to a clearer and more logical manner. A dramatic revision is mandatory for making this manuscript crystal clear for the potential readers. A lot of results with Confidence Intervals are provided without a logical order. The manuscript is extremely confusing for any reviewer highly specialized in statistical methods. For the common reader, reading this manuscript as it has been submitted will cause headaches.

We apologize for the confusion. We have revised the Methods and Results parts to make the manuscript easier to follow.

2) The tables are far away from the text and are not easy to read. Additionally, the legends of these tables should help any common reader to read more easily and understand without any doubt the tables.

We thank the reviewer for the helpful comment. We have revised the Results and Tables notes to make it easy to read.

3) The Material and Methods section is quite short and does not contain any crystal clear explanation justifying the statistical method used.

We thank the reviewer for the comment. We have revised the Material and Methods section to make it clear.

Page 7, lines 143-146: “To account for both the event onset and the time to the event, we calculated hazard ratios (HRs) and 95% confidence intervals (CIs) for the associations of serum retinol (as a continuous or categorical variable) with all-cause mortality using Cox proportional hazards models.”

Page 7-8, lines 151-157: “To minimize the effect of potential confounders on the associations between serum retinol and mortality, we adjusted for age (continuous), sex (female or male), and ethnicity (non-Hispanic white, non-Hispanic black, Mexican American, or other); and subsequently adjusted for education (less than high school, high school or equivalent, or college or above), smoking status (never smoker, former smoker, or current smoker), alcohol consumption (drinker or non-drinker), physical activity (inactive or active), total cholesterol (continuous), and hypertension (no or yes).”

4) Are prediabetes patients receiving any medication or not? Are they following any nutritional diet or not? If they receive any medication or diet, what are these medications or diet(s)?

In the current study, we defined all participants receiving anti-diabetes treatment as diabetes. Therefore, participants who were prediabetes did not self-report using anti-diabetes treatment. Moreover, information about whether they were on a nutritional diet was not available. 

5) Usually Diabetes Mellitus (DM) provokes complications such as the diabetic foot, the Diabetic retinopathy, the Diabetic nephropathy and can provoke strokes and other cardio-vascular complications alone or in association with smoking? This manuscript does not provide the causes of death of ant patient enrolled in this study.

We thank the reviewer for the comment. In this study, we considered all-cause mortality as our outcome, which were derived from the National Death Index up to December 31, 2019. About 44.9% participants with diabetes died from cardiovascular disease death, and 55.1% were due to other causes. We have added this information in the manuscript. 

Methods (Page 6, lines 114-116): “Cardiovascular disease (CVD) death was defined as death from diabetes mellitus (E10-E14), heart disease (codes I00-I09, I11, I13, and I20-I51), or cerebrovascular disease (codes I60-I69). Non-CVD death was defined as death from all other causes.”

Results (Page 9, lines 186-187): “993 participants with prediabetes died (34.1% died from cardiovascular diseases and 65.9% died from non-cardiovascular diseases).” 

Results (Page 9, lines 197-198) “…874 participants with diabetes died (44.9% died from cardiovascular diseases and 55.1% died from non-cardiovascular diseases).”

6) The BMI tends to be replaced by the perimeter of the waist.

We understood the reviewer’s concern. Waist circumference may be more likely to reflect the visceral fat mass than BMI. BMI has some advantages, e.g., it is easier to be measured and has same cut-offs regardless of sex and age. Also, BMI is highly correlated with waist circumference.

Per the reviewer’s We have added a supplementary analysis to adjust for waist circumference instead of BMI. The results were not altered largely (Supplementary Table 1).

Page 8, lines 167: “In the supplementary analyses, we 1) adjusted waist circumference instead of BMI” 

Page 11, lines 224-225 “In the multi-adjusted model, we included waist circumference instead of BMI and the results were not altered largely (Supplementary Table 1).”

7) The duration of DM from its onset to the appearance of complications is a parameter of paramount importance in the prediction of complications including death

We thank the reviewer for the comment. The duration of diabetes per se might be associated with the mortality risk. However, almost half of participants with diabetes (809 missing) did not self-report their diagnosed age, therefore we did not adjust the duration of diabetes in the main analyses. We have added this to the limitation as follows (Page 14, Lines 302-305).

“Finally, although we adjusted a wide range of confounders, there may be residual confounding effects due to not adjusting for potential confounders (e.g., vitamin A supplements, nutritional diet, duration of diabetes, and diabetes complications).”

8) The association of Hypertension with DM can actually lead to severe metabolic syndromes: The manuscript does not evoke how many metabolic syndromes were observed in the cohort studied.

We agreed with the reviewer’s point that the associations between serum retinol and mortality might vary by hypertension. Per the reviewer’s suggestion, we examined the interaction between serum retinol and hypertension for mortality among participants with diabetes by including their interaction term in the model. There was no statistically significant interaction (β coefficient [95% CI]: 0.041 [-0.190, 0.108], P = 0.960), which suggested that the strength of associations between serum retinol and mortality was similar across hypertension status. 

9) No mention of the monitoring of the patients enrolled is provided. How many patients were followed by a cardiologist, by a Nephrologist ? Monitoring the heart of the patients by ultrasonography-Doppler, monitoring the brain of each patient by MRI should have been done in order to evaluate the macro vascular damages caused by DM.

We understood the reviewer’s concern. However, information about whether participants were followed by doctors was not available in this nationally representative study, either for using advanced technologies for monitoring heart or brain. 

10) The rationale for studying retinol is unclear. What is (are) the pathophysiological mechanism(s) susceptible to explain a higher mortality for patients having low concentrations of retinol or high concentrations of retinol in their blood. The answer to this request is of utmost importance for the reviewers and the potential readers.

We thank the reviewer for the comment. We have added more information in Discussion section, to explain the pathophysiological mechanisms supporting the U-shaped association between serum retinol and all-cause mortality.

Pages 13-14, lines 266-290: “There are several plausible pathophysiological mechanisms supporting the U-shaped association between serum retinol and all-cause mortality. Retinol plays an important role in regulating cellular differentiation, immunity and vision (4-6). Moreover, some in vitro studies have reported that lower levels of retinoids (the derivative of retinol) may be associated with increased oxidative stress, inflammation, and endothelial dysfunction (21, 22). Those, in turn, may lead to higher mortality risk. More importantly, we found that the relatively low serum retinol (i.e., <50 μg/dL) was associated with increased risk of mortality in older adults aged ≥60 years. We speculate that aging is especially vulnerable to malnutrition, and more VA is needed to improve the nutritional status in elderly people. On the other hand, there may be a direct acute and chronic toxicity of hypervitaminosis A (23). Elevated retinol levels have been related to embryonic malformations and chronic toxicity, such as ataxia, alopecia, hyperlipidemia, hepatotoxicity, fracture, and visual impairments (24, 25). Furthermore, high retinol levels have been documented as an additional oxidative damage factor (26). In addition, retinol and its derivatives may impact mitochondrial structure and function by causing organelle swelling (27), which could worsen bioenergy disorders, increase oxidative stress, and apoptosis or necrosis. Moreover, serum retinol can bind to retinol-binding protein 4 (RBP4), and is delivered to adipose tissues, further stimulate macrophages bringing about increased local inflammation and systemic insulin resistance through a JNK-dependent pathway (28). Although relatively scarce, some evidence has shown that VA status and its metabolism may contribute to the glucose regulation and fat metabolism (7, 29). Moderate retinoic acid treatment has been shown to reduce blood glucose in Zucker diabetic fatty (ZDF) rats (30), whereas excessive VA intake impaired glucose tolerance and fatty liver in ZDF rats (31, 32). These results may be due to the synergistic interactions of insulin and retinoic acid through the activations retinoid X receptors and retinoic acid receptors signaling pathways (33).”

---

## [Editor Report · Decision Letter 1]

9 Jan 2024

Associations between serum retinol and all-cause mortality among adults with prediabetes and diabetes: A cohort study

PONE-D-23-19123R1

Dear Dr. Guo,

We’re pleased to inform you that your manuscript has been judged scientifically suitable for publication and will be formally accepted for publication once it meets all outstanding technical requirements.

Kind regards,

Billy Morara Tsima, MD MSc

Academic Editor

PLOS ONE
---

## [Editor Report · Acceptance letter]

25 Jan 2024

PONE-D-23-19123R1 

PLOS ONE

Dear Dr. Guo, 

I'm pleased to inform you that your manuscript has been deemed suitable for publication in PLOS ONE. Congratulations! Your manuscript is now being handed over to our production team.

Kind regards, 

on behalf of

Dr. Billy Morara Tsima 

Academic Editor

PLOS ONE